

# A microbial survey of the International Space Station (ISS)

Jenna M. Lang[1], David A. Coil[1], Russell Y. Neches[1], Wendy E. Brown[2,11], Darlene Cavalier[2,3,4], Mark Severance[2,4], Jarrad T. Hampton-Marcell[5,6], Jack A. Gilbert[7,8] and Jonathan A. Eisen[1,9,10]

[1] Genome Center, University of California, Davis, CA, United States of America
[2] Science Cheerleader, United States of America
[3] The Consortium for Science, Policy & Outcomes, Arizona State University, Tempe, AZ, United States of America
[4] Scistarter.org, United States of America
[5] Biosciences Division, Argonne National Laboratory, Lemont, IL, United States of America
[6] Department of Biological Sciences, University of Illinois at Chicago, Chicago, IL, United States of America
[7] Argonne National Laboratory, University of Chicago, Lemont, IL, United States of America
[8] Institute for Genomics and Systems Biology, Argonne National Laboratory, Lemont, IL, United States of America
[9] Evolution and Ecology, University of California Davis, CA, United States of America
[10] Medical Microbiology and Immunology, University of California, Davis, CA, United States of America
[11] Biomedical Engineering, University of California, Davis, CA, United States of America

## ABSTRACT

**Background**. Modern advances in sequencing technology have enabled the census of microbial members of many natural ecosystems. Recently, attention is increasingly being paid to the microbial residents of human-made, built ecosystems, both private (homes) and public (subways, office buildings, and hospitals). Here, we report results of the characterization of the microbial ecology of a singular built environment, the International Space Station (ISS). This ISS sampling involved the collection and microbial analysis (via 16S rRNA gene PCR) of 15 surfaces sampled by swabs onboard the ISS. This sampling was a component of Project MERCCURI (Microbial Ecology Research Combining Citizen and University Researchers on ISS). Learning more about the microbial inhabitants of the "buildings" in which we travel through space will take on increasing importance, as plans for human exploration continue, with the possibility of colonization of other planets and moons.

**Results**. Sterile swabs were used to sample 15 surfaces onboard the ISS. The sites sampled were designed to be analogous to samples collected for (1) the Wildlife of Our Homes project and (2) a study of cell phones and shoes that were concurrently being collected for another component of Project MERCCURI. Sequencing of the 16S rRNA genes amplified from DNA extracted from each swab was used to produce a census of the microbes present on each surface sampled. We compared the microbes found on the ISS swabs to those from both homes on Earth and data from the Human Microbiome Project.

**Conclusions**. While significantly different from homes on Earth and the Human Microbiome Project samples analyzed here, the microbial community composition on the ISS was more similar to home surfaces than to the human microbiome samples. The ISS surfaces are OTU-rich with 1,036–4,294 operational taxonomic units (OTUs

Corresponding author
Jonathan A. Eisen,
jaeisen@ucdavis.edu

per sample). There was no discernible biogeography of microbes on the 15 ISS surfaces, although this may be a reflection of the small sample size we were able to obtain.

## INTRODUCTION

There is a growing appreciation of the importance of microbial communities found in diverse environments from the oceans, to soil, to the insides and outsides of plants and animals. Recently, there has been an expanding focus on the microbial ecology of the "built environment"—human constructed entities like buildings, cars, and trains—places where humans spend a large fraction of their time. One relatively unexplored type of built environment is that found in space. As humans expand their reach into the solar system, with renewed interest in space travel, and with the possibility of the colonization of other planets and moons, it is of critical importance to understand the microbial ecology of the built environments being utilized for such endeavors.

Interest in the microbial occupants of spacecraft long precedes the launch of the International Space Station (ISS) (*Trexler, 1964*; *Silverman, 1971*). Early work primarily focused on ensuring that the surfaces of spacecraft were free of microbial contaminants in an effort to avoid inadvertent panspermia (seeding other planets with microbes from Earth) (*Pierson, 2007*). Work on human-occupied spacecraft such as Mir, Space Shuttles, and Skylab focused more on microbes with possible human health effects. With the launch of the ISS, it was understood that this new built environment would be permanently housing microbes as well as humans. Calls were made for a better understanding of microbial ecology and human-microbe interactions during extended stays in space (*Pierson, 2007*) (*Roberts, Garland & Mills, 2004*; *Ott, Bruce & Pierson, 2004*). Efforts were made to establish a baseline microbial census. For example, *Novikova et al. (2006)* obtained more than 500 samples from the air, potable water, and surfaces of the ISS, over the course of six years.

These early studies were unavoidably limited by their reliance on culturing to identify microbial species. Culture-independent approaches were eventually implemented, including some small-scale 16S rRNA gene PCR surveys (*Castro et al., 2004*), and the Lab-On-a-Chip Application Development Portable Test System (LOCAD-PTS) (*Maule et al., 2009*), which allows astronauts to test surfaces for lipopolysaccharide (LPS— a marker for Gram negative bacteria). Originally launched in 2006, the capability of the LOCAD-PTS was expanded in 2009 to include an assay for fungi (beta-glucan, a fungal cell wall component) and Gram positive bacteria (lipoteichoic acid, a component of the cell wall of Gram positive bacteria.) The first large-scale, culture-independent 16S rRNA gene PCR survey was published only in 2014 using the Roche 454 platform (pyrosequencing), looking at dust on the ISS (*Venkateswaran et al., 2014*). A more recent study examined several samples collected on the Japanese module of the ISS over a period of four years, also sequenced with pyrosequencing (*Ichijo et al., 2016*). We report here on a further effort
involving 16S rRNA gene PCR and sequencing, using the Illumina platform, to examine the microbial communities found on 15 surfaces inside the ISS. The advantage of Illumina sequencing, relative to previous pyrosequencing efforts, is the significant increase in depth of sequencing. This increased depth allowed us to analyze 15–20 times as many sequences as these earlier studies.

The 15 surfaces sampled on the ISS were chosen by the Project MERCCURI team in an effort to make them analogous to (1) the surfaces sampled for the "Wildlife of Our Homes" project (http://homes.yourwildlife.org) (*Dunn et al., 2013a*; *Barberán et al., 2015a*; *Barberán et al., 2015b*), which asked citizen scientists to swab nine surfaces in their homes, and (2) cell phone and shoe swab samples that were also being collected via Project MERCCURI. The sample matching is imperfect; for example, doorsills were used in houses because they collect dust but, in the microgravity of the ISS, dust accumulates in air filters. The motivation for choosing the sites in this way was both to increase public awareness of the microbiology of the built environment, as well as to begin to compare the microbial ecology of homes on Earth with the only current human home in space. We also present a comparison of the ISS swab results with data from 13 human body sites sampled via the Human Microbiome Project. This comparison was done to assess the potential human contribution to the microbial life on the ISS.

We have also compiled a collection of papers on space microbiology in an online resource to provide a more comprehensive historical perspective of this kind of work (see https://www.zotero.org/groups/2008182/spacemicrobiology).

## METHODS

### Surfaces swabbed

Astronauts were asked to swab 15 surfaces on the International Space Station. Below are their verbatim instructions.

(1)  Audio Terminal Unit (telephone) hand held push-to-talk microphone located in the forward portion of the US Lab Module
(2)  Audio Terminal Unit (telephone) hand held push-to-talk microphone located in the aft portion of the US Lab Module
(3)  US Lab Robotic Work Station laptop PC keyboard used to control the robotic arm
(4)  US Lab Robotic Work Station hand controller used to control the movement of the robotic arm
(5)  US Lab Robotic Work Station foothold, left side
(6)  US Lab Robotic Work Station foothold, right side
(7)  One of the main laptop keyboards in the US Lab used to control science experiments and the systems of the space station
(8)  One of the vertical handrails on the equipment racks inside the US Lab
(9)  Air vent in the front portion of the US Lab
(10)  Air vent in the aft portion of the US Lab
(11)  Air vent located on the right crew sleep compartment
(12)  Tab used to open, close, and secure the Nomex privacy panel located on the starboard crew sleep compartment

(13) Air vent located on the port crew sleep compartment

(14) Tab used to open, close, and secure the Nomex privacy panel located on the port crew sleep compartment

(15) Crew Choice Surface: Audio Terminal Unit (telephone) hand held push-to-talk microphone located in the starboard portion of the Harmony module (Node 2).

### Swabbing instructions as given to astronauts

(1) Setup Node-2 Camcorder to capture NanoRacks surface swab Ops throughout the US LAB.

(2) Retrieve a clean NanoRacks Swab Kit. Move to ISS location listed on NanoRacks Swab Kit label.

(3) Remove cotton swab from NanoRacks Swab Kit, being careful not to touch the cotton swab tip to avoid contamination.

(4) Rub cotton swab vigorously against designated surface. Spin and turn the swab to ensure maximum sample collection.

(5) Return cotton swab to NanoRacks Swab Kit and press to close (squeeze excess air out of bag before sealing). Circle number of location swabbed and label with GMT (dd/hh: mm). If swab is contaminated by touching a surface other than the designated location on the label, Label NanoRacks Swab Kit with a large, "X" and move on to the next location. Notify POIC of NanoRacks Swab Kit S/N that was contaminated.

(6) Repeat step 2 to step 6 for all 15 locations listed on the NanoRacks Swab Kit label.

NOTE: An additional large Ziplock Bag is provided (stowed inside the same bag as the NanoRacks Swab Kits) to use per crew preference to separate the used NanoRacks Swab Kits from the clean (unused) NanoRacks Swab Kits for crew efficiency during sampling.

### ISS Crew

Swabbing was conducted during Expedition 39 (http://www.nasa.gov/mission_pages/station/expeditions/expedition39/index.html). The crew included NASA astronauts Steve Swanson and Rick Mastracchio and Russian cosmonauts Oleg Artemyev, Alexander Skvortsov, and Mikhail Tyurin. Japan Aerospace Exploration Agency (JAXA) astronaut Koichi Wakata was the commander for this expedition, and is the astronaut who performed the swabbing.

### Sampling site choice

These surfaces were chosen in an attempt to sample surfaces analogous to those sampled in the pilot study for the Wildlife of Our Homes project (*Dunn et al., 2013a*). For this study, involving 40 homes, volunteers swabbed nine surfaces in their homes: kitchen cutting board, kitchen counter, a shelf inside a refrigerator, toilet seat, pillowcase, exterior handle of the main door into the house, television screen, the upper door trim on the outside surface of an exterior door, and the upper door trim on an interior door. We were not granted access to all corresponding surfaces aboard the ISS. The kitchen surfaces aboard the ISS are in the Russian module, which we did not have permission to access, swabbing the toilet seat was deemed inappropriate due to biosafety concerns, and the exterior surfaces are accessible only via an "Extra-vehicular Activity" (space walk), which was not requested

**Table 1  ISS sample surface descriptions and sequence statistics.**

| Sample | Earth analog | Number of sequences obtained | Number of OTUs observed (at 97% similarity) |
|---|---|---|---|
| Forward lab mic | Cell phone | 45,902 | 1,744 |
| Lab robotic workstation keyboard | None | 31,612 | 1,320 |
| Aft lab mic | Cell phone | 63,958 | 2,457 |
| Lab robotic workstation joystick | Door handle | 76,198 | 2,820 |
| Lab robotic workstation left foothold | Shoe | 77,843 | 1,995 |
| Lab robotic workstation right foothold | Shoe | 74,023 | 2,129 |
| Aft lab vent | Interior door trim | 64,782 | 1,456 |
| Starboard crew vent | Interior door trim | 63,280 | 4,294 |
| Starboard sleep quarters nomex | Pillow | 26,831 | 1,036 |
| Port crew vent | Interior door trim | 50,418 | 1,757 |
| Port sleep quarters nomex | Pillow | 61,306 | 1,349 |
| Node2 mic | Cell phone | 50,416 | 1,429 |
| Lab main keyboard | None | 62,567 | 1,678 |
| Lab handrail | Door handle | 70,418 | 2,904 |
| Forward lab vent | Interior door trim | 57,715 | 2,380 |

for this experiment. We also sought to collect samples that would be analogous to the cell phone and shoe samples that were being obtained from thousands of Citizen Scientists across the country in a different component of Project MERCCURI. A final constraint was the limitation of only 15 swabs that was imposed by NASA, severely limiting the number of replicates we could collect. See Table 1 for a list of the ISS sampling sites and to which Earth samples they were intended to be analogous.

Upon successful completion of the swabbing on May 9, 2014, http://blogs.nasa.gov/stationreport/2014/05/09/iss-daily-summary-report-050914/, all swabs were stored at −80 °C in the Minus Eighty-degree Laboratory Freezer for ISS (MELFI) freezer onboard the ISS, until transfer to the SpaceX Dragon spacecraft. In the Dragon, the swabs were stored at −80 °C in the General Laboratory Active Cryogenic ISS Experiment Refrigerator (GLACIER), that runs off of Dragon's batteries until it is plugged in (either to the ISS or on the ground.) The Dragon re-entered the Earth's atmosphere and splashed down in the Pacific Ocean at 12:05 pm PT on May 18, 2014. Samples were transferred to a cooler with dry ice, and shipped to the Earth Microbiome Project (EMP) lab (http://earthmicrobiome.org) (*Gilbert et al., 2011*).

### DNA extraction and library preparation

All samples were prepared using a modified version of the Mo BIO UltraClean® 96 Well Swab DNA Kit (MO BIO, Carlsbad, CA, USA). Samples were purified using the Zymo ZR-96 DNA Cleanup and Concentrator™-5 kit according to Zymo Protocol (Zymo Research, Irvine, CA, USA). DNA was then amplified using the EMP barcoded primer set, adapted for the Illumina HiSeq2000 and MiSeq by adding nine extra bases in the adapter region of the forward amplification primer that support paired-end sequencing. The V4

region of the 16S rRNA gene (515F-806R) was amplified with region-specific primers that included the Illumina flowcell adapter sequences and a twelve-base barcode sequence. Each 25 µl PCR reaction contained the following: 12 µl of PCR water certified DNA-free (MO BIO), 10 µl of 1× 5 Prime HotMasterMix (5 Prime), 1 µl of Forward Primer (5 µM concentration, 200 pM final), 1 µl of Golay Barcode Tagged Reverse Primer (5 µM concentration, 200 pM final), and 1 µl of template DNA. The conditions for PCR were as follows: 94 °C for 3 min to denature the DNA, with 35 cycles at 94 °C for 45 s, 50 °C for 60 s, and 72 °C for 90 s, with a final extension of 10 min at 72 °C to ensure complete amplification. Amplicons were quantified using PicoGreen (Invitrogen, Carlsbad, CA, USA) and a plate reader. Once quantified, different volumes of each of the products were pooled into a single tube so that each amplicon was represented equally. This pool was then cleaned up using UltraClean® PCR Clean-Up Kit (MO BIO, Carlsbad, CA, USA), and then quantified using Qubit (Invitrogen, Carlsbad, CA, USA). Sequencing of the prepared library was performed on the Illumina MiSeq platform, using the sequencing primers and procedures described in the supplementary methods of *Caporaso et al. (2012)*.

## Bioinformatic analysis

Unless otherwise noted, all microbial community analyses were conducted using the QIIME workflow version 1.8 or R (*R Core Team, 2014*). All Python scripts referred to are components of QIIME (*Caporaso et al., 2010*).

Demultiplex and QC: an in-house script was used to assign sequences to samples, using dual-index barcoding. This script is available on GitHub (https://github.com/gjospin/Demul_trim_prep). This script allows for one base pair difference per barcode. The paired reads were then aligned and a consensus was computed using FLASH (*Magoc & Salzberg, 2011*) with maximum overlap of 120 and a minimum overlap of 70 (other parameters were left as default). The custom script automatically demultiplexes the data into fastq files, executes FLASH, and parses its results to reformat the sequences with appropriate naming conventions for QIIME v. 1.8.0 in fasta format.

OTU assignment and QC: Chimeric sequences were identified using usearch61 as implemented in the identify_chimeric_seqs.py script, resulting in the removal of 8,760 sequences. The pick_open_reference_otus.py script was used to cluster sequences at 97% similarity to generate OTUs (Operational Taxonomic Units). Taxonomy was assigned to each OTU by comparing a representative sequence from each cluster to the gg_13_8_otus reference taxonomy provided by the Greengenes Database Consortium (http://greengenes.secondgenome.com) (*McDonald et al., 2011*). OTUs that were classified as chloroplasts or mitochondria were removed from further analysis. The number of high-quality sequences remaining per sample ranged from 26,831 to 77,843 (see Table 1). All subsequent beta diversity analyses (comparisons across samples) were performed with all samples rarefied to 26,830 sequences.

## Comparison of ISS surfaces to analogous surfaces in homes on Earth and to the Human Microbiome Project

The sequences and associated metadata from a 40-home pilot study for the Wildlife of Our Homes Project are available for download from Figshare (*Dunn et al., 2013b*).

We also obtained 100 samples from each of 13 body sites from the HMP Data Portal (http://hmpdacc.org/HM16STR/) (*Huttenhower et al., 2012*; *Gevers et al., 2012*). These two additional datasets were used in a combined analysis with the ISS sequences presented here. Because the sequences from the three projects are not all the same lengths, each dataset was independently analyzed using a closed-reference OTU-picking approach, with a 97% similarity cutoff, and the resultant biom tables were merged with the merge_otu_tables.py script. While the closed-reference approach will miss any novel taxa, this was required since both of our comparison datasets were analyzed this way. To account for uneven sampling depth, all samples in the combined analysis were rarefied to 1, 000 sequences. Shannon diversity, as well as non-metric multidimensional scaling (NMDS) based on Bray–Curtis (*Bray & Curtis, 1957*) and Unweighted Unifrac (*Lozupone & Knight, 2005*) distances were computed and plotted using Phyloseq (*McMurdie & Holmes, 2013*) and the ggplot2 (*Wickham, 2009*) packages in R (*R Core Team, 2014*).

### Comparison to rooms with mechanical ventilation or open windows

We obtained a list of human pathogens, compiled by *Kembel et al. (2012)* from the author. We then used BLAST (*Altschul et al., 1990*) to search a representative sequence from each of the ISS OTUs against the NCBI Reference Sequence (RefSeq) database (*Pruitt, 2004*). OTUs with 97% similarity to an organism that was on the list of known pathogens were flagged as "related to a known human pathogen". The phylogenetic diversity (Faith's PD) was calculated using the alpha_diversity.py script, with samples rarefied to 700 sequences.

## RESULTS AND DISCUSSION

### Overall taxonomic diversity of ISS surfaces and comparison to previous high-throughput 16S rRNA gene study

After filtering chimeric and eukaryotic sequences from the data, the number of sequences per surface sampled ranged from 26,221–76,656. Open-reference clustering at 97% similarity resulted in 12,554 OTUs. This exceeds the number observed by *Venkateswaran et al. (2014)* and *Ichijo et al. (2016)* which is not surprising, given the increased sampling depth in this study (~1 million versus ~50,000–71,000 high-quality sequences.) Our study also had four notable, qualitative differences from these earlier studies. In *Venkateswaran et al., (2014)*, more than 90% of all sequences were assigned to four bacterial genera (*Corynebacterium*, *Propionibacterium*, *Staphylococcus*, and *Streptococcus*), while in the study here, they comprised only 24% of the data (9.6%, 0.05%, 10.7%, and 3.6%, respectively). *Ichijo et al. (2016)* did not report genus level taxonomy but the phyla containing these groups (Firmicutes and Actinobacteria) were highly abundant in all samples. Second, Venkateswaran et al. found no evidence of archaea in their samples, even when interrogating with archaeal-specific primers, but we did find evidence for a very low relative abundance archaeal presence (2,335 sequences, from three archaeal phyla). No archaeal results were reported by *Ichijo et al. (2016)*. Next, despite the fact that Venkateswaran et al. were able to culture many spore-forming organisms from their samples, they observed no sequence data from putative spore-forming organisms. However, a large percentage of sequences in our study are from genera containing known spore-forming genera: 20.9% *Bacillus*

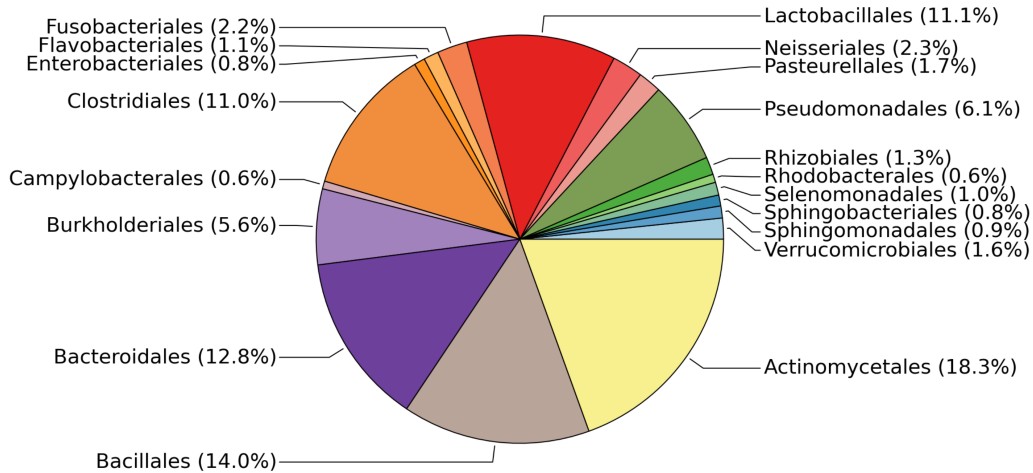

**Figure 1** **Relative abundances of the most common bacterial families found on surfaces of the ISS.** Pie chart showing the relative abundances of the most common bacterial families found on the 15 surfaces of the International Space Station. This graph was produced using METAGENassist (*Arndt et al., 2012*).

and 9.6% *Clostridium*. These differences are potentially due to differences in PCR primers and/or DNA extraction method, both of which have known taxonomic biases (*Brooks et al., 2015*). Lastly *Ichijo et al. (2016)* noted a significant amount of both Legionellaceae and Neisseriaceae which are both of potential concerns as families containing may pathogenic members. However, our study observed no OTUs for these groups which is most likely due to sampling site differences or PCR primer differences as noted above.

The 19 most abundant orders found in our study represent 93.8% of the data (Fig. 1). Within each of these 19 orders, the most abundant genus found in our samples tends to be human-associated (Table 2). This is not surprising, as the only source of microbial influx is via occasional crew and cargo deliveries aboard spacecraft that have been stringently cleaned to avoid microbial contamination. It should be noted, as with all 16S rRNA gene surveys, that nothing can be said about the viability of these bacteria. Typically much of the bacterial DNA on a surface is from dead or non-viable organisms. In built environments on earth this DNA is assumed to come from many sources including outdoor air, soil, and the passage of people and animals. On the ISS all of these taxa, viable or not, represent organisms that have managed to survive the various protocols designed to limit them, the most likely passage being on the crew themselves.

There were no apparent biogeographical patterns on the ISS surfaces. That is, there were no significant differences between samples obtained from the different modules (crew vs lab) or different surface types (keyboards, vents, or handheld mics). This can be visualized in Fig. 2, in which each point represents one of the 15 samples, and the distance between samples indicates the overall difference in community composition. In Fig. 2A, the metric used to calculate the distance between samples is the Bray–Curtis dissimilarity, and in Fig. 2B, an alternative distance metric (Unifrac) is used, which takes into account the phylogenetic distance between the OTUs in samples. For the most part, all 15 samples form a tight cluster on the NMDS plots, but there is one sample, the starboard crew

**Table 2 The most abundant organisms on the ISS are human-associated.** From each of the 19 orders shown in Fig. 1, we selected the most abundant genus and conducted a literature review to identify whether or not it is known to occur in association with the human microbiome.

| Order | % abundance | Dominant Genus | Common habitat | Reference |
|---|---|---|---|---|
| Actinomycetales | 18.3 | *Corynebacterium* | Human skin, oral cavity | *Grice et al. (2009), Zaura et al. (2009)* |
| Bacillales | 14 | *Staphylococcus* | Human skin, oral cavity | *Grice et al. (2009), Zaura et al. (2009)* |
| Bacteroidales | 12.8 | Unclassified Rikenellaceae/S24-7 | Animal gut | *Langille et al. (2014), Krych et al. (2015)* |
| Lactobacillales | 11.1 | *Streptococcus* | Human oral cavity | *Aas et al. (2005)* |
| Clostridiales | 11 | *Finegoldia* | Human skin | *Higaki & Morohashi (2003)* |
| Pseudomonadales | 6.1 | *Pseudomonas* | Human skin | *Cogen, Nizet & Gallo (2008)* |
| Burkholderiales | 5.6 | Unclassified Comamonadaceae | Environmental | *Willems (2014)* |
| Neisseriales | 2.3 | *Neisseria* | Human mucous membranes | *Liu, Tang & Exley (2015)* |
| Fusobacteriales | 2.2 | *Fusobacterium* | Human oral cavity | *Schwarzberg et al. (2014)* |
| Pasteurellales | 1.7 | *Haemophilus* | Human respiratory tract | *Murphy et al. (2007)* |
| Verrucomicrobiales | 1.6 | *Akkermansia* | Human gut | *Belzer & De Vos (2012)* |
| Flavobacteriales | 1.1 | *Capnocytophaga* | Human oral cavity | *Zaura et al. (2009)* |
| Selenomonadales | 1 | *Selenomonas* | Human oral cavity | *Ribeiro et al. (2011)* |
| Sphingomonadales | 0.9 | *Sphingomonas* | Environmental | *Seifried, Wichels & Gerdts (2015)* |
| Sphingobacteriales | 0.8 | Unclassified Sphingobacteriales | Environmental | *Steyn et al. (1998)* |
| Enterobacteriales | 0.8 | Unclassified Enterobacteraceae | Animal gut | *Linton & Hinton (1988)* |
| Rhizobiales | 0.6 | *Methylobacterium* | Environmental | *Knief et al. (2010)* |
| Campylobacterales | 0.6 | *Campylobacter* | Animal gut | *Young, Davis & DiRita (2007)* |

vent, that appears distinct from all of the other samples in Fig. 2A. In Fig. 2B, that same sample, as well as the aft lab vent sample appear separate from the others. In order to visualize which OTUs are contributing the most to the uniqueness of those samples, we looked at the overall distribution of the most abundant bacterial families in those samples. The three most abundant families in the starboard crew vent sample are Bacteroidaceae, Ruminococcaceae, and Verrumicrobiaceae (comprising 60.1% of all sequences); and the three most abundant families in the aft lab vent sample are Rikenellaceae, Bacteroidales S24-7, and Lactobacillaceae (comprising 60% of all sequences). In Fig. 3, the relative abundance of these six families in all 15 samples from the ISS provides a clear indication that they are driving the distinctiveness of those two samples.

The massive increase in environmental 16S rRNA gene surveys over the last several years has seen a greater understanding of the caveats and limitations with this kind of data, in parallel with their unambiguous utility in understanding microbial communities. When this experiment was designed in 2012, negative kit controls were not common but now they are considered standard for good reason (*Salter et al., 2014*). Lacking a kit control, we cannot say for certain which low-level taxa may have come from the swabs or reagents used themselves.

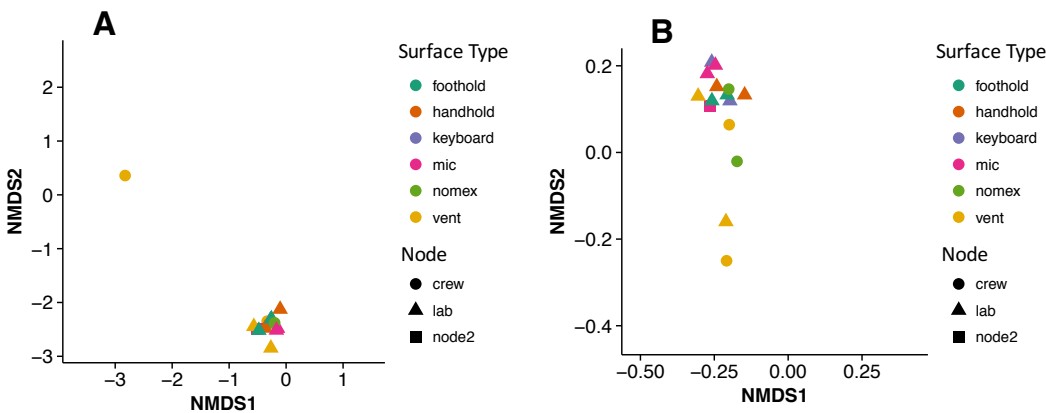

**Figure 2** **Non-metric multidimensional scaling (NMDS) ordination plots of 15 ISS surface samples.** Non-metric multidimensional scaling (NMDS) ordination plots, based on Bray–Curtis (A) or Unweighted Unifrac (B) distances between the samples obtained from the International Space Station. In these plots, points that are closer together have more similar microbial communities. Note, there is a (starboard) crew vent sample that does not cluster with the other ISS samples in (A) and in (B), a second sample (aft lab vent) appears closer to it. This graph was produced using the Phyloseq package (*McMurdie & Holmes, 2013*) in R (*R Core Team, 2014*).

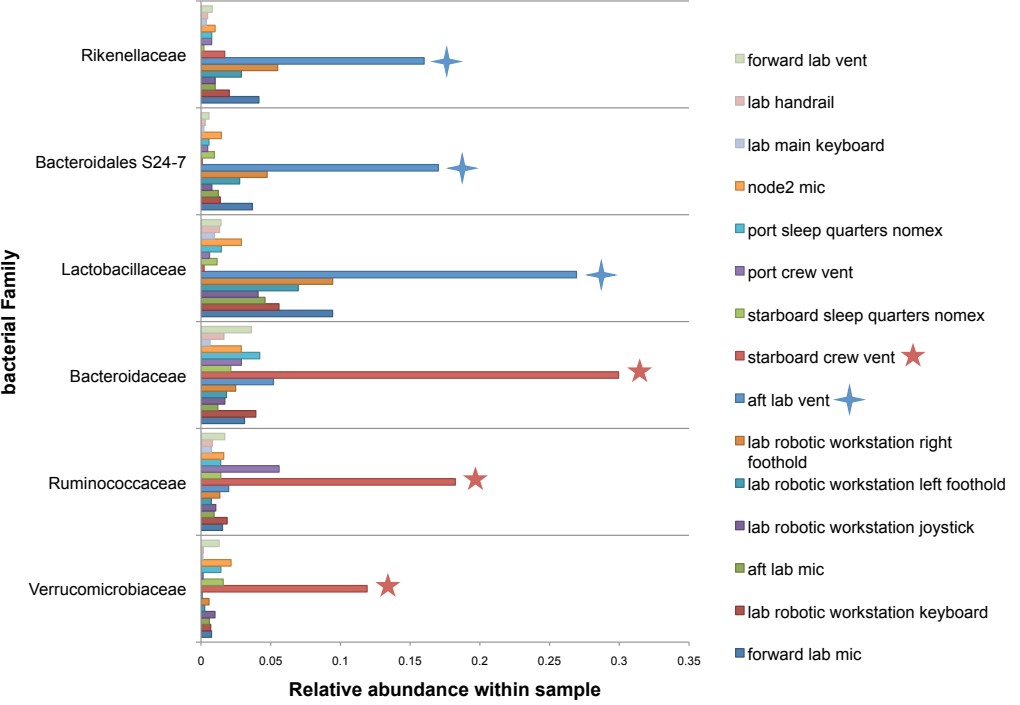

**Figure 3** **Most abundant bacterial families found in each of the two "outlier" samples on the ISS.** Bar chart showing the distribution across all samples of the 3 most abun- dant bacterial families found in each of the two samples (starboard crew vent and aft lab vent) that do not cluster with the others in Fig. 2. All six of these families are known to be found in association with human (or animal) gastrointestinal tract.

Peer J

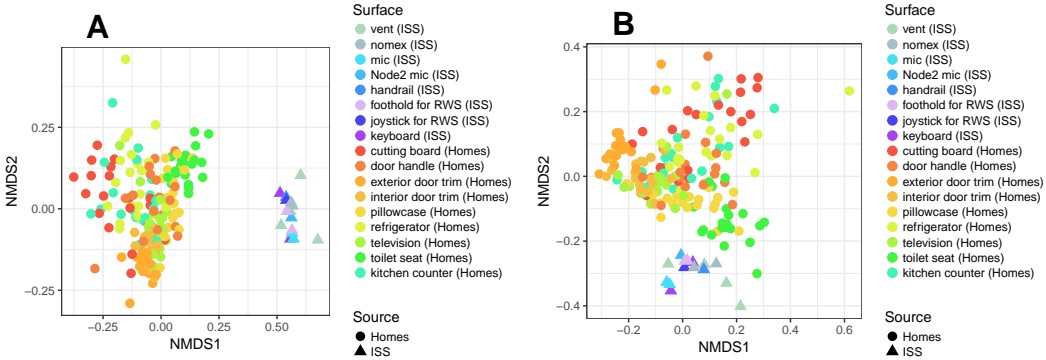

**Figure 4** **Non-metric multidimensional scaling (NMDS) ordination plots of ISS surface samples.** Non-metric multidimensional scaling (NMDS) ordination plots, based on Bray–Curtis (A) or Unweighted Unifrac (B) distances between samples obtained from the International Space Station and samples obtained from homes on Earth. In these plots, points that are closer together have more similar microbial communities. We found that the ISS samples and Earth home samples were significantly different from each other, both based on the Bray–Curtis dissimilarity (adonis, $R^2 = 0.0666$, $P = 0.001$) and the Unifrac distance (adonis, $R^2 = 0.04189$, $P = 0.001$). Note, the crew and lab vent samples that are distinct from the other ISS samples Fig. 2, do not cluster with any of the Earth home surfaces. This graph was produced using the Phyloseq package (*McMurdie & Holmes, 2013*) in R (*R Core Team, 2014*).

## Comparison to the microbial communities of homes on Earth and from the Human Microbiome Project

To put the microbial communities that we found on ISS surfaces in the context of homes on Earth, we compared them to the communities found in samples taken by citizen scientists when they swabbed nine surfaces throughout 40 homes, as part of the ''Wildlife of Our Homes'' project (*Dunn et al., 2013a*). We found that the ISS and homes on Earth were significantly different from each other, both based on the Bray-Curtis dissimilarity (adonis, $R^2 = 0.0666$, $P = 0.001$) and the Unifrac distance (adonis, $R^2 = 0.04189$, $P = 0.001$). These differences can be visualized in the ordination plots in Figs. 4A and 4B.

It is perhaps not surprising that the insular environment of the ISS would be unlike homes on Earth. Unlike the ISS, homes on Earth are exposed to a variety of sources of microbes, including the outdoor air, tracked-in soil, plants, pets, and human inhabitants (*Barberán et al., 2015a*; *Barberán et al., 2015b*). The dominant source of microbes on the ISS is presumably the human microbiome. All spacecraft and cargo undergo rigorous decontamination procedures before launch to rendezvous with the ISS. Therefore, we hypothesized that the microbial communities of the ISS surfaces might be more similar to human-associated microbial communities than Earth home surfaces. To test this hypothesis, we obtained 16S rRNA gene sequence data for 100 random samples from each of 13 body sites from the HMP Data Portal (http://hmpdacc.org/HM16STR/) (*Huttenhower et al., 2012*) (*Gevers et al., 2012*). The microbial communities associated with the ISS, homes on Earth, and the HMP samples were significantly different from each other (adonis, $R^2 = 0.08$, $P < 0.001$) (Also see Fig. 5). We note that as with any meta-analysis, this difference could be also be partly due to differences in sample collection/preparation. However, the ISS communities are significantly more similar to the Earth home samples

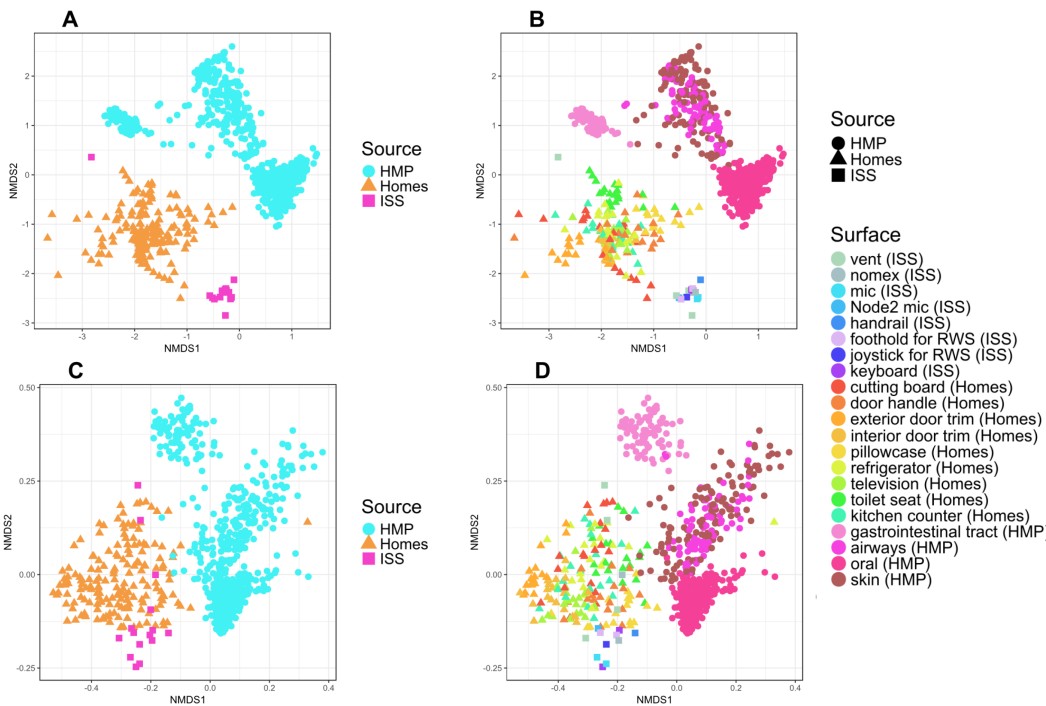

**Figure 5  NMDS plots showing clustering of ISS, Earth homes, and Human Microbiome Project body sites.** Non-metric multidimensional scaling (NMDS) ordination plots, based on Bray–Curtis (A and C) or Unweighted Unifrac (B and D) distances between samples obtained from the International Space Station, from homes on Earth, and from 13 body site from the Human Microbiome Project. The plots in (A) and (B) show identical data, as do the plots in (C) and (D). The points in (A) vs. (B) and (C) vs. (D) are colored differently as an aid for visualization. In these plots, points that are closer together have more similar microbial communities. The microbial communities associated with the ISS, homes on Earth, and the HMP samples were significantly different from each other (adonis, $R^2 = 0.08$, $P < 0.001$). Note: the crew and lab vent samples that are distinct from the other ISS samples in Fig. 2 are more similar to the human gastrointestinal tract samples from the HMP. This graph was produced using the Phyloseq package (McMurdie & Holmes, 2013) in R (R Core Team, 2014).

than the HMP samples (Student's $t$-test, $p < 0.00001$). This combined analysis also indicates that the starboard crew vent sample, which appears quite distinct from the rest of the ISS samples in Fig. 2A, is more similar to the human gastrointestinal HMP samples, which is corroborated by the dominance of animal gut-related OTUs found in that sample (see Fig. 3, and Table 2).

Finally, because the ISS is designed only to house six crew members, for a stay of six months each, only 220 individuals have visited the ISS since the year 2000. We hypothesized that there might be a relatively low microbial diversity on the ISS, either due to having a few total number of OTUs, or due to the dominance of a very few species. In Fig. 6, we note that Shannon diversity (which takes into account both the number of OTUs present, and how evenly our sequences are distributed throughout those OTUs) is actually relatively high on the ISS.

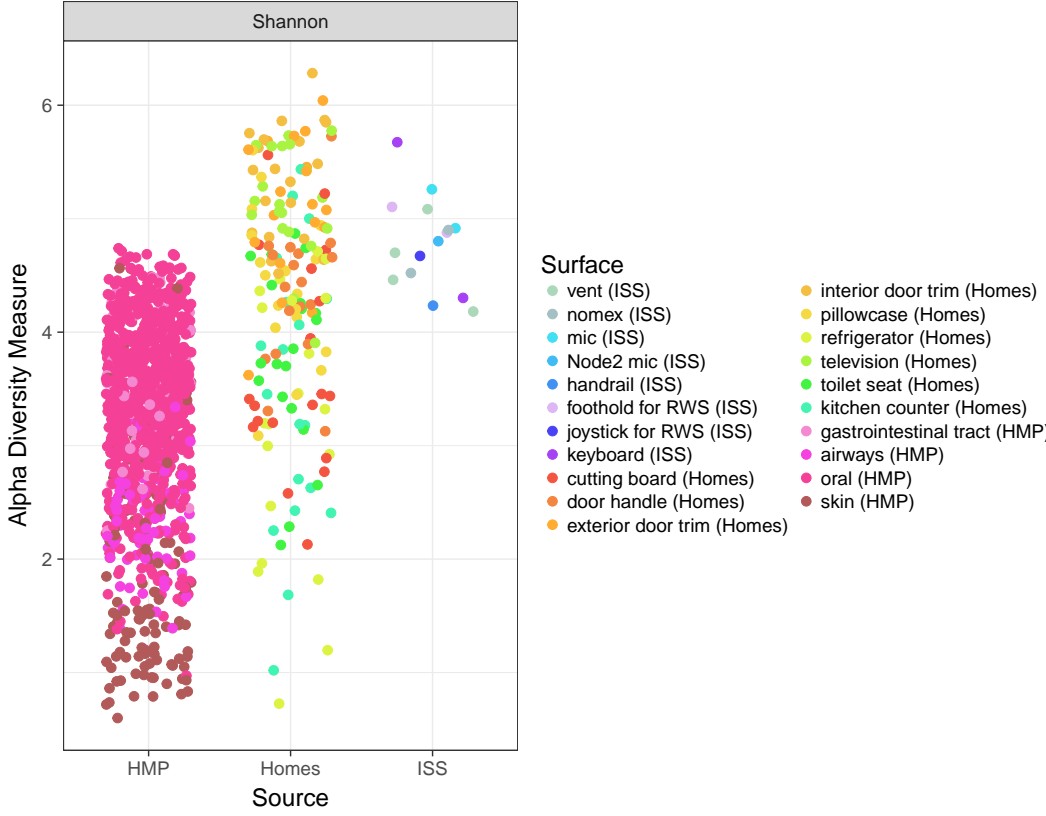

**Figure 6** **Comparison of Shannon diversity among the ISS, Earth homes, and HMP body sites.** Shannon diversity, a measure of how many OTUs there are as well as how evenly the counts of individuals are distributed across OTUs is plotted for every sample. There is wide variation among the HMP samples, with the oral (blue) and gastrointestinal (green) samples typically having more diversity than the skin (pink) or airway (coral) samples. Surfaces on the International Space Station have relatively high Shannon diversity, on par with that of the most diverse HMP samples, and the average home sample. This graph was produced using the Phyloseq package (*McMurdie & Holmes, 2013*) in R (*R Core Team, 2014*).

## Comparison to rooms with mechanical ventilation or open windows

*Kembel et al. (2012)*, showed that rooms in a health-care facility that were primarily ventilated via an open window had greater phylogenetic diversity and lower proportion of OTUs closely related to known human pathogens than rooms that were mechanically ventilated. The only window on the ISS is never opened, and the doors are opened only briefly, every few months. Therefore, we hypothesized that for the samples from the ISS, the phylogenetic diversity would be lower and the proportion of OTUs closely related to known human pathogens would be higher than that seen for mechanically ventilated rooms. To test this hypothesis, we obtained the list of known human pathogens compiled by *Kembel et al. (2012)*, and followed their procedure to identify the proportion of OTUs in the ISS samples that were closely related to them (see Methods for details). Surprisingly, but reassuringly, we found that the ISS samples are similar in both phylogenetic diversity and the proportion of OTUs closely related to known human pathogens as compared to
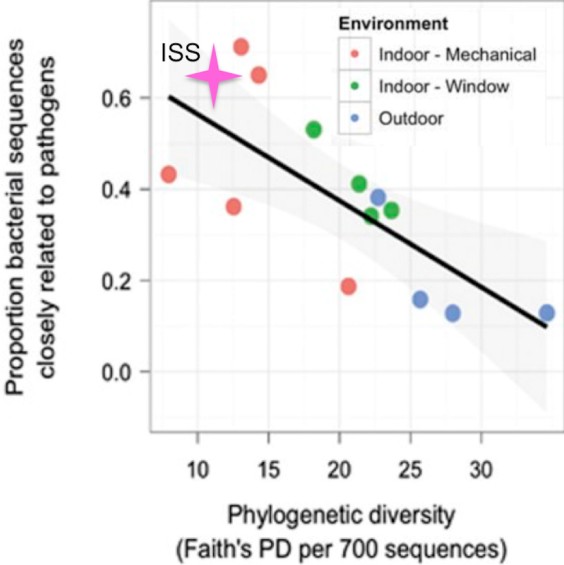

**Figure 7** **Proportion of OTUs found in the ISS samples that were closely related (97% sequence similarity) to human pathogens versus the phylogenetic diversity of those samples.** This figure was modified from Fig. 4A of (*Kembel et al., 2012*). The pink star represents the ISS samples. The plot shows the proportion of OTUs that were closely related (97% sequence similarity) to human pathogens versus the phylogenetic diversity of those samples.

the mechanically ventilated rooms in the health-care facility (Fig. 7). As with the studies above, some observed variance may be due to differences in sample collection/preparation.

## CONCLUSION

This is the first time that the ISS has been analyzed in the broader context of the "microbiology of the built environment", and is the most in-depth comparison of the microbial communities found on the ISS to those found either in buildings or in the human microbiome. Perhaps surprisingly, given the extreme rarity of exchange with any external microbes, we found the ISS to be OTU-rich, and more similar to the surfaces of human homes on Earth than it is to human bodies. We found that the ISS is home to at least 12,554 distinct microbial OTUs, including Archaea in very low abundance, and that the proportion of OTUs that are closely related to known human pathogens is on par with similar built environments on Earth. Given the low number of samples in this study, no viability assessment, as well as the lack of sample preparation control we view these results as simply a starting place for more detailed future studies.

As outlined in the 2010 US National Space Policy and in the bipartisan NASA Authorization Act of 2010, NASA is targeting the 2030s for a manned spaceflight to Mars, with one ultimate goal of having people live and work on the Martian surface (see http://www.nasa.gov/exploration and http://www.nasa.gov/mars). We know that the microbial communities found in our terrestrial built environments play an important role in human health. Therefore it is crucial to characterize and understand the microbial

population of the only environment in which people are currently living and working in space. This study is one small step in that direction.

## ACKNOWLEDGEMENTS

The authors would like to thank Summer Williams for the inception of the idea to get Science Cheerleader involved in space research. In addition we give thanks to Carl Carruthers at Nanoracks LLC for managing our space payload. We are also grateful to Holly Menninger and Rob Dunn for sharing data from the Wildlife of Our Homes pilot project, and Steven Kembel (ORCID ID: 0000-0001-5224-0952) for publishing the original version of Fig. 7 in an open access journal, as well as sharing the underlying data.

### Funding

This work was supported by the Space Florida ISS Research Competition (http://www.spaceflorida.gov/iss-research-competition), http://SciStarter.com, and a grant to Jonathan A. Eisen from the Alfred P. Sloan Foundation as part of the ''Microbiology of the Built Environment'' program. The funders had no role in study design, data collection and analysis, decision to publish, or preparation of the manuscript.

### Grant Disclosures

The following grant information was disclosed by the authors:
Space Florida ISS Research Competition.
Alfred P. Sloan Foundation.

### Competing Interests

Jonathan A. Eisen is an Academic Editor for PeerJ.

### Author Contributions

- Jenna M. Lang conceived and designed the experiments, analyzed the data, wrote the paper, prepared figures and/or tables, reviewed drafts of the paper.
- David A. Coil, Russell Y. Neches and Wendy E. Brown conceived and designed the experiments, reviewed drafts of the paper.
- Darlene Cavalier, Mark Severance and Jonathan A. Eisen conceived and designed the experiments, contributed reagents/materials/analysis tools, reviewed drafts of the paper.
- Jarrad T. Hampton-Marcell performed the experiments.
- Jack A. Gilbert performed the experiments, contributed reagents/materials/analysis tools.

### Data Availability

Sequencing data is deposited at NCBI under BioProject PRJNA376404. All data and analysis files are available at: Lang, Jenna (2017): A microbial survey of the International Space Station (ISS). figshare. https://doi.org/10.6084/m9.figshare.4244123.v3.

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
