# Peer review of "A microbial survey of the International Space Station (ISS)"

_PeerJ, doi:10.7717/peerj.4029_

## Round 0.1 · original submission · Minor Revisions

Please address comments by both reviewers in a point by point rebuttal letter.

Reviewer 2 has declared their potential COI with J Eisen, but we are satisfied that their review is unbiased.

Reviewer 1 ·

Basic reporting

The submitted manuscript reports the microbial assemblage of the International Space Station (ISS), and compares the ISS assemblage to microbiomes of homes and humans. The manuscript is generally well-written with somewhat sufficient and relevant references to support their claims. However, given the very low number of samples (which is in itself understandable because of the sampling site and circumstances), one may expect more rigorous analyses to be done to further describe the ISS assemblage based on the limited number of samples. For example, the manuscript can add the following analyses/discussions to make up for the limited number of samples included:

1.1 The authors have compared their findings with the previous work by Venkateswaran et al. (2014), but do not compare their findings with that of Ichiro et al. (2016).

1.2 Given that the crew vent sample appears to differ from the majority of other samples, would be interesting to investigate this sample further at the sub-genus/strain/species level.

1.3 Greater efforts will need to be addressed to the figures. For example, in Figure 2, the title of the legend groups appear to lack consistency in capitalization ("Surface_Type" but "node"). Also, there should not be underline in the figure legend "Surface_Type." In Figure 4, it is extremely hard to visualize the different sample groups, using the current colour labelling scheme (which is very gradient-like). The authors may want to manually fix the colour for each sample group. Also, the list of sample groups currently are in alphabetical order, but this order does not help in the overall understanding of the ecology. Alternatively, sample groups can be listed by their built environment group (e.g. all sample groups within Homes listed first, then sample groups within ISS, or vice versa). Figure 5 Panels B and D, and Figure 6 suffer from the same weakness.

Experimental design

2.1 The Methods section is described in detail.

2.2 It is implied that the authors have filled the knowledge gap by using the Illumina technology to increase the sequencing depth to explore the ISS microbiome in greater detail. Perhaps strengthening that point in the introduction and conclusion will add impact to the manuscript.

2.3 The analyses described appear to have done well, however additional analyses should be performed to strengthen the impact of the work given the very low sample number.

2.4 The lack of controls in assessing for potentially contaminating taxa is a great weakness of this work, but the authors have explained the reason for not having them. However, it would still be beneficial to include controls that the lab group has subsequently produced after the start of this work, to show readers that the authors made an effort to account for potential contamination during sample prep and sequencing. If possible, this is highly recommended. If not, please explain why this is not possible.

Validity of the findings

3.1 Given the low number of samples, it is necessarily to indicate in discussion that the lack of replicates may affect the validity of the results.

3.2 There is no indication where the sequences of the ISS samples can be accessed. Please make that available in a public repository.

3.3 While the authors indicate that the findings will have in space exploration in the future, as most readers will not be heading to space in the immediate future, can readers learn anything about the built environment on Earth from this ISS study?

Additional comments

Line 39: change "Earth homes" to "homes on Earth"

Line 93: change from "example" to either "exemplify" or "assess"

Line 185: twelve base change to "twelve-base"

Section "Comparison of ISS surfaces in homes on Earth and to the Human Microbiome Project:" besides the point that surfaces on the ISS were selected to mirror that selected for a previous home study, please indicate more succinctly the rationale for comparing the microbiomes of the ISS to homes. Is this because humans will potentially living in the space in the future? If anything, it is my humble opinion that the ISS environment may more resemble that of cleanrooms. Unfortunately, cleanroom environments are not described in this study, and a simple taxonomic or community analysis should have been described. Also, as any differences in methodology can potentially affect sequence-based microbial community surveying results, please indicate whether these different studies adopted identical sample collection, preparation, and sequencing methods. If not, add a disclaimer that method differences may have affected the comparison of the studies.

Line 272: outside air change to "outdoor air"

Line 311: suggest to remove double negative in sentence.

Line 326-330. Is there an ecological/rational reason for the discrepant community found in the crew vent sample, and its resemblance to human gut communities?

Please also fix the in-text citation and references throughout. Additional proof-reading of the manuscript will be beneficial to minimize typos (random hash-tags appearing in manuscript, mathematical power symbol represented as ^ instead of a superscripted 2, etc.). Refer to https://peerj.com/about/author-instructions/ for more information.

·

Basic reporting

No comment.

Experimental design

No comment.

Validity of the findings

No comment.

Additional comments

Lang et al. present a manuscript that examines the microbiome of the ‘built environment’ of the International Space Station (ISS) using 16S rRNA sequencing on the Illumina MiSeq. Sample diversity did not cluster by sample surface. Comparisons of ISS swabs to data from homes and humans revealed that the surfaces of human homes are more similar to ISS surfaces than human body surfaces are, but that neither is significantly similar. The proportion of known human pathogens on the ISS surfaces was comparable to those in ventilated rooms in a health-care facility.

Overall this is a clearly-worded, concise manuscript. The question of which microbes inhabit the surfaces of human-built environments is both important and interesting, especially in the case of the ISS. Many of the comments/edits that I have suggested are minor, with the exception of a few bigger concerns that I would like to address. Firstly, in the introduction where the authors mention studies that also used deep sequencing to examine ISS microbiome, it would be helpful to have a sentence or two expanding on the results that they obtained, in order to compare your results. You did have a comparison in the results section to the Venkateswaran study, so you could expand upon the Novikova and Castro studies there. At least a comparison of the surface samples that are relevant to your study. While I understand the necessity of closed OTU picking for the comparison of your data to HMP and Homes data, it would be good to include a statement about the limitations of this method. Also, you are losing a lot of your data by rarefying to 1000 or less sequences for your comparisons with HMP, Homes and mechanically ventilated areas. Could you please explain why it was necessary to rarefy to such a level? Finally, you should put the statistical numbers stated in the text in their appropriate figures too, for example the Adonis tests. Below are some smaller edits with reference to their location in the text:

- Line 32 in the Abstract – “as plans for human exploration and colonization...come to fruition.” I think it was phrased better in line 53 of the introduction where it says “possibility of colonization”. It comes off a little strong when stated as a fact.
- As stated above I think it would be worth expanding on the findings of previous studies in paragraphs 2 and 3 of the introduction
- Line 97 This link leads to a site that directs you to a new link. So maybe just put the new link?
- Line 165 – for Table 1 it might be good to have an explanation in the supplemental information on why you correlated particular Homes samples with particular ISS samples. Most of them are intuitive but for example the similarities between air vents and door trimming is not obvious
- Line 181 – Should the EMP primers have a citation?
- Line 192 – This is a little vague – do you mean there were equal concentrations of each product in the pooled sample?
- Line 199 – Is there a link for the QIIME workflow?
- Lines 201 and 209 – there are some ### and I’m not sure if some information is missing there. The sentence in 201 is definitely not complete.
- Line 228 Describe limitations of closed OTU reference picking
- Line 232 – you are missing a space after the bracket
- Line 235 – Did you mean rarefy to “1000 sequences”
- Line 243 – Why rarefy to 700 sequences here instead of using the 1000 cutoff from before
- Line 299 – Missing a space before the bracket

---

## Round 0.2 · accepted · Accept

The manuscript has now been sufficiently improved for publication.